# Dysregulated Immune Responses in SARS-CoV-2-Infected Patients: A Comprehensive Overview

**DOI:** 10.3390/v14051082

**Published:** 2022-05-18

**Authors:** Igor Kudryavtsev, Artem Rubinstein, Alexey Golovkin, Olga Kalinina, Kirill Vasilyev, Larisa Rudenko, Irina Isakova-Sivak

**Affiliations:** 1Institute of Experimental Medicine, 197022 Saint Petersburg, Russia; igorek1981@yandex.ru (I.K.); arrubin6@mail.ru (A.R.); kirillv5@yandex.ru (K.V.); vaccine@mail.ru (L.R.); 2Almazov National Medical Research Centre, 197341 Saint Petersburg, Russia; golovkin_a@mail.ru (A.G.); olgakalinina@mail.ru (O.K.)

**Keywords:** COVID-19, antigen-presenting cell, Th cell subsets, CD8+ T cell, monocyte, Th17, follicular Th cell, cellular immunity, humoral immunity, post-COVID-19 syndrome

## Abstract

Severe acute respiratory syndrome coronavirus 2 (SARS-CoV-2) was first detected in humans more than two years ago and caused an unprecedented socio-economic burden on all countries around the world. Since then, numerous studies have attempted to identify various mechanisms involved in the alterations of innate and adaptive immunity in COVID-19 patients, with the ultimate goal of finding ways to correct pathological changes and improve disease outcomes. State-of-the-art research methods made it possible to establish precise molecular mechanisms which the new virus uses to trigger multisystem inflammatory syndrome and evade host antiviral immune responses. In this review, we present a comprehensive analysis of published data that provide insight into pathological changes in T and B cell subsets and their phenotypes, accompanying the acute phase of the SARS-CoV-2 infection. This knowledge might help reveal new biomarkers that can be utilized to recognize case severity early as well as to provide additional objective information on the effective formation of SARS-CoV-2-specific immunity and predict long-term complications of COVID-19, including a large variety of symptoms termed the ‘post-COVID-19 syndrome’.

## 1. Introduction

The SARS-CoV-2 virus is known for employing a variety of molecular mechanisms to evade antiviral innate and adaptive host immunity, in particular, those aimed at inhibiting interferon production, dendritic cell (DC) activation, and adaptive immune priming [1]. Many additional factors have been identified that may influence the severity of COVID-19, including the age and sex of SARS-CoV-2-infected individuals [2]; various comorbidities such as diabetes, hypertension, obesity, heart and kidney disease, and malignancy [3]; specific genotypes of class I and class II major histocompatibility complex molecules [4]; and single-nucleotide variants in genes encoding important transcription factors, cytokines, pattern recognition receptor, etc. [5]. High levels of circulating autoantibodies have also been associated with a severe respiratory disease that activates the complement cascade, neutrophil stimulation, and thrombosis [6].

COVID-19 is also associated with a dysregulation of multiple biological pathways. The primary issue is abnormal activation of different cell types and persistent pro-inflammatory cytokine production in severe cases of the disease, and these two factors usually converge, inducing critical disturbance on both local and systemic levels. For instance, ‘cytokine storm’, a widespread and massive release of pro-inflammatory cytokines and chemokines, induces extrapulmonary involvement and possibly fatal complications associated with multisystem inflammatory syndrome leading to alterations in differentiation, proliferation, and activation of various immune cells, as well as in the ‘polarization’ of CD4+ and CD8+ T cells [7,8]. Furthermore, high plasma concentrations of extracellular vesicles, originating from the cytoplasmic membrane budding in activated platelets and endothelial cells [9], leukocytes and neutrophils [10], or even virus-specific T and B cell subsets [11], contribute to the pathological inflammatory response during the acute COVID-19. Thus, the absence or low level of the SARS-CoV-2-specific T and B cells in the peripheral blood of patients with COVID-19 could be caused by defective induction of adaptive immunity and could result in greater disease severity.

In contrast, the presence of T cells and antibodies is associated with the successful resolution of COVID-19, suggesting that T and B cell responses are important for controlling virus spreading and the successful resolution of the primary SARS-CoV-2 infection [12]. SARS-CoV-2-specific CD4+ T cells can be detected as early as on days 2–4 post-symptom onset, and rapid induction of functionally effective SARS-CoV-2-specific T cells is associated with prompt viral clearance and lower disease severity, while the delayed appearance of IFNγ-secreting cell is shown for patients with severe COVID-19 [13,14]. Thus, T cell response seems to develop early and correlate with protection, but it is relatively weakened in severe disease and is associated with intense activation and lymphopenia [15]. Although a high rate (up to 90%) of antibody seroconversion is detected among SARS-CoV-2-infected individuals 7–14 days post-symptom onset [16,17], the impairment of antiviral T cells functions is not compensated effectively by the increased antibody production and is considered as one of the main immunity-related causes of death from COVID-19 [18].

Thus, the clinical spectrum of COVID-19 is notably broad, ranging from asymptomatic to mild, moderate, severe, and critical forms. It highlights the importance of analyzing the changes in T and B cell subsets and their phenotypes during the acute phase of SARS-CoV-2 infection to reveal new biomarkers for early recognition of the case severity. These biomarkers can potentially provide additional objective information on the effective formation of SARS-CoV-2-specific immunity or predict long-term complications of COVID-19, including a large variety of symptoms termed the ‘post-COVID-19 syndrome’.

## 2. Antigen-Presenting Cells in COVID-19

The innate immune system functions as the first line of host defense against pathogens, including SARS-CoV-2, and antigen-presenting cells play a critical role in triggering adaptive immune responses mediated by antigen-specific T and B cells [19]. The ability to present antigens is possessed by DCs of various subsets delivering antigens from peripheral tissues to lymphoid tissue, B-lymphocytes that capture peripheral antigens from the lymph, and macrophages that capture and present antigens in the connective tissue of various localization [20].

### 2.1. Dendritic Cells and Their Subsets

A decrease in the total pool of circulating DCs is characteristic not only of the acute period of SARS-CoV-2 but is also found in recovered patients [21,22]. Peripheral blood DCs are a heterogeneous population of leukocytes which is traditionally subdivided into myeloid or conventional CD123–CD11c+ dendritic cells (cDCs) and CD123+CD11c− plasmacytoid dendritic cells (pDCs) [23,24]. In turn, cDCs are usually divided into two main subpopulations of cDC1 and cDC2, which differ in both their phenotype and functions [25].

During SARS-CoV-2 infection, DCs are the leading producers of main pro-inflammatory cytokines, and they can release type I and type III IFNs, which help to achieve a sustained antiviral state that limits the viral spread and suppress the viral infection [26]. A decrease in circulating myeloid (CD11c+CD123lo/-) and plasmacytoid (CD11c–CD123+) DCs was noted in severely ill COVID-19 patients [27,28,29] (Figure 1). Another study shows that an increase in the cDC/pDC ratio in peripheral blood can be considered a promising marker for the severe course of COVID-19 [21].

Further studies revealed significant alterations in the phenotypic and functional characteristics of various DC subpopulations [29] (Figure 1). In all patients with COVID-19, the expression level of CD45RA on the surface of circulating pDCs was reduced regardless of the disease severity, while patients with a severe form of the disease demonstrated a decrease in HLA-DQA2 mRNA and HLA-DR at the protein level. The analysis of the cDC subpopulation composition showed that the proportion of CD5+ DC2s and immature DC progenitor cells increases in patients with moderate severity of COVID-19, and the proportion of CD163–CD14– DC3 decreases during the severe course of the disease. A detailed analysis of the phenotype of various DC subpopulations in patients with severe COVID-19 revealed a decrease in HLA-DR and CD86 levels in all cell populations, except cDC1 [29]. In response to in vitro stimulation with ligands for TLR3, TLR4, TLR7, or TLR8, the key populations of DCs from COVID-19 patients (i.e., pDC, cDC1, and cDC2) expressed less CD80, CD86, CCR7, and HLA-DR than the cells from corresponding populations of apparently healthy volunteers [21]. It should be noted that the decrease in the functional activity of circulating DCs can be quite long-lasting: the reduced CD86 expression was commonly observed in recovered patients as well, although it took much less time to restore the normal density of HLA-DR and CCR2 [21]. In addition, it was shown that the increased expression of CX3CR1 on pDCs and cDC1s was associated with disease severity [32]. Furthermore, the persistent downregulation of CD86 and upregulation of PD-L1 expression in cDCs correlated with COVID-19 severity and could be associated with a reduced capacity to activate ‘naïve’ and memory Th cells in peripheral lymphoid tissue [33].

As the literary data demonstrated, infection with the SARS-CoV virus in vitro was accompanied by a decrease in the production of type I IFN and MHC class I on DCs and fibroblasts [34]. Similarly, pDC from patients with severe COVID-19 showed a decrease in the expression of pS6, mTOR, and some other genes responsible for the regulation of type I IFN expression [35]. Perhaps this underlies the weaker production of type I IFN and TNF by the pDCs in the SARS-CoV-2-infected patients after in vitro stimulation. Such a dramatic change in the functional activity of the virus-infected DCs may be associated with the ability of SARS-CoV-2 to suppress phosphorylation of the STAT1 transcription factor, which is involved in the regulation of signaling pathways of type I, type II, and type III IFNs [36].

Overall, the decrease in circulating DCs may be due to their migration into the lymphoid tissue, where these cells perform the functions related to initiating the adaptive antiviral immune response (Figure 1). Furthermore, the lower levels of DC activation, even in comparison with milder forms of COVID-19, indicate that SARS-CoV-2 uses highly effective strategies to evade the immune responses. Finally, the accumulation of various immature DC precursors in peripheral blood (possibly due to their release from the red bone marrow) may also negatively affect the efficiency of antigen-specific responses since they do not yet possess effector properties and cannot stimulate T cells. Interestingly, SARS-CoV-2-infected patients show low frequencies in some circulating DC subsets (CD1c+ cDCs and pDCs) and alterations in DCs homing (CCR7 and integrin β7) and activation (CD86 and PD-L1) cell surface antigens, which are not restored for more than 7 months after the acute phase of infection independently of previous hospitalization [37].

### 2.2. Monocytes and Macrophages

A reduced number of CD14+ monocytes relative to control values was observed in patients with a severe course of COVID-19 [38]. The expression density of the CD64 activation marker on the surface of monocytes remarkably increased during COVID-19, reaching its maximum in patients with mild disease. As the severity of the disease increased, the CD64 level decreased, reaching the minimum values in severe forms of COVID-19. An increase in the proportion of Ki-67+ monocytes was closely related to the amount of CRP, IL-6, and chemokines MCP-1, IP-10, and IL-10, which have the highest values in critically ill patients [38].

Another study of COVID-19 patients found a negative correlation between the serum concentration of IL-6 and the level of HLA-DR expression on the CD14+ monocytes, as well as between the total number of lymphocytes and the absolute number of HLA-DR mRNA isolated from CD14+ monocytes [39] (Figure 2). An increase in atypical CD14+CD16+ monocytes with a higher expression of CD80 and CD206 and a capability to secrete larger amounts of IL-6, IL-10, and TNFα than CD14+CD16+ monocytes of normal size was noted in the blood of infected people [40].

When stimulated in vitro, the monocytes from COVID-19 patients synthesized significantly lower levels of pro-inflammatory cytokines IL-1β, TNFα, IL-6, and MCP-1, compared to the cells obtained from apparently healthy donors [35]. Controversially, another study indicated that in severe COVID-19 cases, peripheral blood monocytes significantly increased expression of IL-6, TNF, IL-1β, and their receptors, as well as pro-inflammatory chemokines including CCL2, CCL3, and CCL4 [41].

Under physiological conditions, about 85% of the total monocyte population in circulation are cells with the CD14^High^CD16^Low^HLA-DR^High^ phenotype which, upon infection, rapidly leave the bloodstream and migrate to inflamed peripheral tissues [42]. Patients with COVID-19 showed an increase in the population of intermediate monocytes with the CD14+CD16+ phenotype, which was generally the highest in patients with a mild disease outcome [38,43] (Figure 2). The severe course of COVID-19 could be closely related to the growth of the classical monocytes’ population in the peripheral blood, as well as a decrease in the intermediate and nonclassical monocytes [44]. Moreover, reduced surface expression of the co-stimulatory CD86 and the antigen-presenting HLA-DR was noted for all three populations, with the most pronounced differences observed for the intermediate monocytes of COVID-19 patients and those of the control group [43]. There was also an increase in the expression of the activation marker CD169 on classical and intermediate monocytes in all patients infected with SARS-CoV-2 [45], whose level was closely associated with the appearance of CD64, CD68, and CD38 on the cell surface, as well as with the decrease in CD86 and HLA-DR. With increasing disease severity, a shift in the phenotype of classical monocytes towards a lower expression of HLA-DR, CD33, and CD11c was found [46].

The hallmarks of severe COVID-19 forms are decreased levels of HLA-DR and CD86, as well as increased levels of CD163 in all subsets of circulating monocytes. The authors also noted an increase in the receptor for thrombomodulin CD141 on nonclassical and intermediate monocytes in severe COVID-19 cases [29].

The study by Silvin et al. reported that intermediate monocytes (CD14^high^CD16^high^) increased in patients with mild COVID-19, compared to healthy individuals and subjects with severe disease, while a distinctive feature of the monocyte pool of severely ill patients was a drop in the nonclassical monocyte population (CD14^low^CD16^high^), which allowed for distinguishing these patients from the other groups [47,48]. However, these results were not confirmed by other researchers, who failed to find any significant differences between the subset composition of monocytes in patients with COVID-19 and the control group [27].

Further proteome and transcriptome analyses of monocytes in patients with mild disease revealed an increase in the proportion of activated classical monocytes with the CD14+HLA-DR^hi^CD11^hi^ phenotype (HLA-DRA^hi^CD83^hi^), which were absent in both the severe COVID-19 and control groups [45]. It should be noted that such significant differences in the subpopulation composition of monocytes can be closely related to the time after infection with SARS-CoV-2, since intermediate and nonclassical monocytes can disappear from the circulation at the initial stages of infection, whereas, at later stages, their number could increase significantly, especially in severely ill patients [45].

The severe cases of COVID-19 were associated with the appearance of monocytes expressing a low level of cell-surface HLA-DR and possessing an anti-inflammatory profile, which manifested in the active expression of genes encoding the MAFB transcription factor, PLBD1 (phospholipase B domain containing 1) and CD163 [48]. The lower expression of HLA-DR on monocytes, along with an elevated proportion of such monocytes in circulation, was considered to be a poor prognostic marker indicating the impaired functional activity of monocytes. For example, in septic conditions, the increased level of HLA-DR^lo^ monocytes in circulation is associated with poor disease outcomes [49]. In addition, abnormalities in the production of pro-inflammatory cytokines, including the release of IL-1β, were noted in severe COVID-19 cases [48], which also indicated the impairment of the monocyte’s ability to generate an effective inflammatory response. Another feature of the production of the regulatory molecules in severe forms compared with the mild course of COVID-19 is a high expression of type I IFN by classic monocytes, combined with the production of the key pro-inflammatory cytokines of the TNF and IL-1 families [44].

At the same time, a reduced level of HLA-DR expression (or the CD14+HLA-DR^–/lo^ phenotype) is characteristic of myeloid-derived suppressor cells (MDSC), a heterogeneous and highly specialized subpopulation of monocytes possessing pronounced anti-inflammatory properties [50]. Based on the decreased expression of MHC class II molecules and the increased level of calprotectin, Xu et al. found an elevated proportion of MDSC in the peripheral blood of the patients with severe COVID-19 compared to those with mild disease and healthy volunteers [51]. The level of these MDSC-like monocytes positively correlated with the levels of C-reactive protein (CRP) and IL-6, as well as with the neutrophil/lymphocyte ratio. Similar results were obtained by other researchers who noted a rapid elevation in the proportion of HLA-DR^lo^CD14+ monocytes in patients with severe COVID-19 [29,47]. These findings indicate that the expansion of circulating MDSC-like monocytes possessing immunosuppressive properties in the severe course of COVID-19 is closely associated with an elevated concentration of calprotectin in circulation, as well as with low levels of nonclassical monocytes in the peripheral blood.

Therefore, the alterations caused by the SARS-CoV-2 infection are associated with a change in the number of monocytes in peripheral blood, as well as in their subset composition (primarily the reduction in “nonclassical” monocytes) and functional activity (the impaired ability to present antigens and co-stimulate) (Figure 2). Finally, in post-COVID-19 patients, the numbers of intermediate monocytes were lower than in acute patients but still significantly higher compared to healthy controls; post-COVID-19 monocytes also showed the highest expression of CCR2, CCR5, CD86, and HLA-DR together with the lowest expression of CD11b when compared to healthy controls [52].

### 2.3. B-lymphocytes

Relative to control values, the number of peripheral B cells in COVID-19 patients was reduced [38,53,54,55]. This decrease was especially noticeable in severely ill patients compared to mild-to-moderate cases. The patients experienced an increase in the proportion of Ki-67+ cells among the general population of CD19+ lymphocytes whose level correlated with CRP serum concentration. The infection was also associated with an elevated proportion of circulating precursors of plasma cells with the CD27^hi^CD38^hi^CD24− phenotype, while the assessment of IgD and CD27 expression by circulating B cells showed the expansion of the double-negative (DN) IgD−CD27− cells, whose number positively correlated with the disease severity (Figure 3). As for memory B cells, the level of CD27+IgD+IgM+ cells in patients with COVID-19 was significantly lower than the control values [38].

The decreased concentration of B cells was associated with a decline in both ‘naïve’ B cells and switched and unswitched memory B cells [53]. On the other hand, the number of IgM+ and IgM– plasmablasts and DN memory B-cells (CD27–IgD–) was significantly higher in the SARS-CoV-2-infected patients. Moreover, within this population, patients with severe disease had an increased proportion of IgD–CD27–CXCR5– B cells that neither enter the B cell area nor participate in the development of a secondary response upon antigen re-exposure [54]. The impaired differentiation of B cells was also confirmed by the reduction of ‘naive’ IgD+CD27− cells, transitional IgD+CD27−CD10+CD45RB− cells, and follicular CXCR5+ (IgD+CD27−CD10−CD73+) cells in the peripheral blood of patients with severe COVID-19, as compared to convalescent COVID-19 patients and healthy controls [54]. However, when B-lymphocytes were activated in vitro, the cells of patients with COVID-19 did not significantly differ in their proliferative activity from apparently healthy volunteers (e.g., in the proliferation index and the number of dividing cells per cycle). At the same time, IgM+ and switched memory B cells gradually increased in most patients, particularly two weeks after the disease onset, while the proportion of circulating plasmablasts decreased, reaching average values by the day of discharge from the hospital [55] (Figure 3).

The elevated number of DN B cells in circulation in patients with COVID-19 may indicate extra-follicular mechanisms for the development of a specific humoral response, which may dominate in patients with a severe course of the disease [54,56]. Apparently, there is hyperactivation of B cells, which is manifested in the expansion of CD11c+CD21− DN2 cells and the precursors of plasma cells with the CD27+CD38^hi^ phenotype and represents a poor prognostic sign [56]. The circulating plasmablasts were characterized by high expression of Ki67, although there was little or no CXCR5 on their surface, which is responsible for migration to B cell areas of peripheral lymphoid organs [27]. In COVID-19, circulating CD27+CD38+CD138+ B cells not only contained high levels of Ki67 in the cytoplasm but also expressed the activation marker CD95 on their membrane, which could indicate a recent migration of B cells from the germinal centers of lymphoid tissue [57]. At the same time, a significantly higher proportion of circulating CD19+CD20–CD38^high^CD27^high^ plasmablasts was a common characteristic of all COVID-19 patients, regardless of the disease severity [55,58,59]. However, RBD-specific circulating plasmablasts were found in sufficient quantities even at the acute stage of infection [60].

On the other hand, the proportion of memory B cells in circulation was significantly reduced in patients with COVID-19, regardless of the disease severity [61]. In severe and critically ill patients, the number of IgD+CD27+ memory B cells decreased, while a significantly reduced population of IgD–CD27+ memory B cells was noted only in severe patients when compared to healthy donors. These observations have been confirmed by other studies [53,54,57,58], which further indicate that the processes of maturation and formation of memory B-cells occurring in peripheral lymphoid organs are significantly impaired by the infection. Moreover, a deeper analysis of the key B cell subsets demonstrated that high numbers of CD27+CD38^hi^, CD21–CD11c– DN3, and IgD–CD27+ memory B-cells were common in critically ill patients, whereas those with severe disease were characterized by increased numbers of CD21–CD11c+ DN2, atypical CD27-negative memory cells, and CD11c+IgD+CD38–/+ activated ‘naive’ cells [61] (Figure 3).

Another sign indicating impaired maturation and differentiation of effector B cells is the release of CD21-negative B-lymphocytes into the blood [27,56]. Thus, in patients with mild and severe COVID-19, the number of CD21+CD27+ cells in circulation reduced relative to the control, while the proportion of CD21–CD27– cells significantly increased [27]. It should be noted that CD21 plays an important role in the activation of B cells; therefore, its lower expression on the cell membrane may be closely related to the impairment of B cells functional activity [62]. More recently, increased levels of both plasma cell precursors (CD19+CD138+CD38+) and functionally defective CD21– B cells have been observed in patients with sepsis, which also indicates a severe infectious process [63]. On the other hand, CD21^low^ B lymphocytes can be considered cells that have just left the germinal center and are precursors of plasma cells [64]. In this case, the accumulation of CD21-negative B cells in the blood may be closely associated with active maturation processes in lymphoid tissues. This is partially supported by the observed increase in the proportion of these cells in patients with mild COVID-19 compared with critically ill patients and the control group [56] and can be considered a good prognostic sign.

Another sign of B cell malfunction in COVID-19 is a sharp decrease in the expression of CXCR5, a chemokine receptor responsible for the migration of B cells to B cell areas of peripheral lymphoid organs [54,57] (Figure 3). Moreover, for COVID-19 convalescents, lower expression levels of CXCR5 on IgD+CD27+ and IgD–CD27+ memory B cells were also noted, which indicates a durable impairment in B cell function during the period of recovery.

As for the antigen-specific B lymphocytes, the cells capable of recognizing RBD were found among double-negative B cells, IgD–CD27+ memory B cells, and circulating plasmablasts [54]. The authors suggested that such distribution of cells may indicate that the SARS-CoV-2-specific humoral response associated with antibody class switching can occur outside the germinal center or ‘extrafollicularly’, which might prevent the establishment of long-term immunological memory. Three months after infection, RBD-specific B lymphocytes had the phenotype of mature class-switched memory B cells (IgD–CD27+), although there were CD24– cells in circulation, as well as many marginal zone B cells with the CD27+IgD+IgM+ phenotype, which are produced from the germinal center-independent pathway [65]. These findings also indirectly indicate an extrafollicular B cell response in COVID-19.

At the same time, recent studies have found that S-specific memory B-cells with the IgD–CD21+CD27+CD38–CD71^int/-^ phenotype are detected in patients who recovered from severe COVID-19 six months ago, suggesting that effective immunity can be induced even in this group of patients [66]. On the other hand, the level of RBD-specific memory B cells remains unchanged for at least six months after infection with SARS-CoV-2 [67]. Moreover, a change in clonality and a higher affinity of the B-cell receptor has been noted in circulating memory B-cells, which indicates the processes of somatic hypermutation occurring in lymphoid tissues and further evolution of antigen-binding sites of an antibody for a long period after recovery. Similarly, other authors found the maintenance of RBD-specific memory B cells for at least six months, even when serum neutralizing antibody titers significantly declined [68]. These data provide further evidence that SARS-CoV-2-specific humoral response can be effectively induced even if the composition of B cells is significantly impaired or the morphology of B cell areas in peripheral lymphoid tissues is altered, which will be discussed below.

## 3. Structural Impairment of Lymphoid Tissue in COVID-19

### 3.1. Spleen

Significant SARS-CoV-2-induced changes were observed when examining spleen samples. A case analysis of a 55-year-old patient with COVID-19 revealed that along with the loss of follicles in the white pulp of the spleen, there was lymphoid hyperplasia and a mosaic structure of the white pulp, i.e., some lymphoid follicles had a normal intact structure, while most of them lacked germinal centers, and in some cases, there was an enlarged mantle zone but no clear marginal zone [69]. At dissection, postmortem spleens of COVID-19 patients were generally contracted with shrunk capsules and were characterized by blood clots, anemic infarction, and hemorrhages [70]. Moreover, atrophic changes of the white pulp (with fewer lymphoid follicles) and the enlargement of the red pulp were noted in these spleens. The cellular composition of the spleens substantially changed as well, including a decline in CD20+ B-lymphocytes, CD3+ T-lymphocytes, and proliferating Ki67+ cells, with an elevated number of macrophages. Throughout the entire period of the disease, the decreased numbers of Tfh cells in the CD4+ICOS+ and CD4+CXCR5+ phenotypes were noted both in the lymph nodes and in the spleen, with CD4+Bcl-6+ Tfh cells being nearly absent in comparison with the control [54]. Apart from the decrease in the proportion of Tfh cells in lymphoid tissues (both in the spleen and in the lymph nodes), the number of Th2 cells also reduced, whereas the levels of Th1 and Th17 increased relative to the control.

The white pulp of COVID-19 patients was characterized by the reduction in both the relative count and the volume of lymphoid follicles as well as the decreased number of germinal centers containing Bcl-6+ B lymphocytes [54]. At the same time, the distribution of follicular dendritic cells (FDC) in the B cell areas did not change and was close to the average levels of apparently healthy patients. The authors suggested that such significant changes in the lymphoid tissue could result from the overproduction of TNFα by the cells that were part of the follicle or surrounding it or from a generally high level of this cytokine circulating in patients with acute COVID-19. Indeed, the role of TNFα in the reorganization of lymphoid tissue during acute infectious processes has been recently demonstrated in a murine model [71,72]. It has been also noted that the peripheral lymphoid organs of patients with COVID-19 contain an increased proportion of IgD−CD27− B cells and IgG-expressing plasmablasts, which are found both within and outside the follicles, indicating the predominance of an extra-follicular B-cell immune response.

### 3.2. Lymph Nodes

Significant changes in the morphological characteristics of the microvascular architecture were observed in the lungs and other organs and tissues of patients with severe forms of COVID-19 [73]. Thus, the pulmonary pathobiology of COVID-19 was characterized by severe endothelial injury (associated with intracellular localization of SARS-CoV-2 and disrupted cell membranes), alveolar capillary microthrombi, and the signs of intussusceptive angiogenesis leading to the deformation of vessels. Such remarkable changes in the microvasculature can not only affect the lymphoid tissue’s architecture, which is responsible for the maturation and differentiation of cells, but they can also influence the migration of effector immunocompetent cells into inflamed peripheral tissues.

There are numerous studies devoted to the analysis of morphological changes in lymph nodes of various localization (pulmonary hilar lymph nodes, mediastinal, paratracheal, paraesophageal, paraaortic) collected during an autopsy. Several studies reported the impaired organization of B cell areas, caused mainly by the reduction in the volume of germinal centers or their complete absence [54,70]. There was a significant decrease in the number of cells that form germinal centers of lymph nodes, including follicular dendritic cells, Bcl-6+ follicular T-helper cells, and B-lymphocytes. Similar results were obtained in the analysis of the pulmonary hilar lymph nodes of COVID-19 patients: the proportion of PD-1+ Tfh and the volume of germinal centers in B cell areas decreased [74]. Duan et al. also noted that CD20+ B cells were distributed diffusely throughout the tissue, and CD138+ plasma cells could be located both within the preserved germinal centers and in the lymph node parenchyma. In addition, diffusely distributed B cells expressing activation-induced cytidine deaminase (AID) persisted in the lymph nodes of patients with severe COVID-19, which indicated the ability of at least some B cells to receive stimulating signals from Tfh and trigger the generation of high-affinity antibodies [54]. In contrast, some studies did not report abnormalities in the structure of B cell follicles of the lymph nodes [75,76], although Hanley et al. noted a relative depletion in the paracortical areas of the lymph nodes, whereas plasma cells were detected in the medullary areas.

The analysis of lymph node samples collected from 12 COVID-19 cases revealed the enlargement of the subcapsular sinuses that were infiltrated by CD68+ macrophages, although the proportion of CD3+ T-lymphocytes and CD20+ B-lymphocytes in the lymph node cortex did not change [70]. The infection resulted in a higher number of cells in the lymphoid tissue that died due to apoptosis or necrosis [77], which could be directly related to the reduced level of lymphocytes in the peripheral blood. Furthermore, some authors noted a rise in the proportion of phagocytic histiocytes (macrophages) in the sinuses of the hilar lymph nodes, while the total lymphocyte count decreased in patients with severe COVID-19 [76]. These data further indicate massive cell death in the lymphoid tissue in severe disease with a fatal outcome.

Overall, the published data indicate a significant impairment of the architectonics and cell composition of peripheral lymphoid organs, which might be closely related to the development of specific immune responses mediated by T and B cells. Abnormalities in the structure of B cell areas are easily detected by morphological assessment, whereas the development of T-cell immune responses can only be assessed by indirect evidence.

## 4. CD4 T Cells and Their Subsets in COVID-19

Within the total pool of peripheral blood leukocytes, the T cell count diminishes in COVID-19 patients depending on the severity of the disease, and a higher proportion of these cells could be considered a good prognostic sign [38]. The level of circulating T cells was lower in COVID-19 patients than in the control group or COVID-19 convalescents [57]. Moreover, the number of T cells in the peripheral blood was inversely related to the concentrations of IL-6 and IL-10, which significantly increased with the progression of the disease severity [78]. The significantly decreased number of CD3+ T cells, as well as CD3+CD4+ and CD3+CD8+ T cells, dynamically correlated with the severity of COVID-19 cases [38,57,79]. Impaired thymus function and decreased ‘naïve’ T cell thymic output in patients with acute COVID-19 were confirmed by the low levels of peripheral blood circular excision products, so-called TRECs (T cell receptor excision circles), in patients with severe and critical COVID-19 [80,81]. In addition, a correlation was observed between the level of T cells in the peripheral blood and the severity of COVID-19 in APACHE III scores [27]. Moreover, the authors observed inverse correlations between the level of T-helpers and the concentrations of D-dimer, ferritin, and CRP in the serum of COVID-19 patients [57]. Decreased CD3+CD4+ T cell count at the time of hospital admission was closely associated with a poor prognosis of COVID-19 and the development of severe bilateral pneumonia [82]. The authors suggested that the level of CD3+CD4+ T cells could have a predictive value for the early detection of severe COVID-19 forms and the selection of patients requiring rapid aggressive treatment with corticosteroids or IL-6 inhibitors. However, IL-6 blockade may not be effective in some severe COVID-19 [83], and largely depended on many factors including baseline IL-6 levels, PaO_2_/FIO_2_, requirement of HFO or NIV, levels of CRP, ferritin, D-dimer, and LDH [84,85].

### 4.1. Maturation, Differentiation, and Activation of CD4+ T-lymphocytes

The analysis of T-helper differentiation in peripheral blood revealed a relative loss of ‘naïve’ CD4 T cells (with CD45RA+CD27+CCR7+CD95− phenotype) but a significant increase in the amount of EM2 (CD45RA−CD27−CCR7+) and EMRA (CD45RA+CD27−CCR7−) cells in COVID-19 patients compared to controls [57]. Lower circulating CD27+CD28+CD45RA+CCR7+ ‘naïve’ CD4+ T cell counts predicted a poor prognosis in patients with acute COVID-19 [86] (Figure 4). Similar results were obtained by other researchers who found a significant increase in mature effector T-helpers with the CCR7−CD45RA+ CD28−CD27+/− phenotype in patients with COVID-19 [53]. However, a study by Spoerl et al. revealed an elevated proportion of CM T-helpers with exclusively the CCR7+CD45RO+ phenotype in patients with severe COVID-19 compared with mild disease and healthy volunteers [87]. These results were confirmed by the observation that CM T-helpers dominated in the peripheral blood, whereas the T-helpers of the effector memory subsets EM2 and EM3 dominated in BALF of COVID-19 patients [88]. In contrast, cytotoxic T cells in BALF samples were represented by resident memory cells, EM4, and TEMRA.

As for the activation markers, the level of CD38 and HLA-DR expression increased on T-helpers of the peripheral blood of COVID-19 patients, with the maximum values of CD38+HLA-DR+ cells found in severe cases [87] (Figure 4). For all non-naive CD4 T-cell subsets (primarily for EM1, EM2, and EM3), an increased number of both CD38+HLA-DR+ and also Ki-67+ cells were found in patients with COVID-19 [57]. The number of these subsets in the peripheral blood positively correlated with the concentration of ferritin and APACHE III scores, which further indicated the association between the level of T-helper activation and the disease severity [57]. Furthermore, the expression level of HLA-DR and CD38 on CD4+ and CD8+ T cells in peripheral blood significantly increased in patients with COVID-19, and the maximum values were found in patients with a poor disease outcome [88] (Figure 4).

Another study found an association between CD4+ T cells expressing programmed cell death protein-1 (PD-1 or CD279) and APACHE III scores in patients with COVID-19 [27]. In addition, the expression of co-inhibitory receptors BTLA and TIGIT was higher on CD4+ T cells, while the number of the cells positive for the ecto-5′-nucleotidase CD73, involved in the modulation of innate immune activation during the viral immune response, reduced relative to the control [89]. The levels of HLA-DR+CD4+ and PD-1+CD4+T cells (along with the concentrations of troponin I, CRP, D-dimer, and LDH) can also serve as predictors of a poor disease outcome [90]. Apart from the increased proportion of PD-1+ T-helpers, there was an accumulation of CTLA-4+ cells whose level correlated with the disease severity [87]. Upon in vitro stimulation with anti-CD3/CD28 antibodies, CD4 T cells of COVID-19 patients showed a lower capacity to express IFNγ and IL-2 and proliferated together with the number of CFSE^low^ cells (a measure of T-cell proliferation) compared to similar T-cell subsets of the control group [21]. Moreover, patients with COVID-19 had significantly less central and effector memory CD4 T-cell subsets capable of IFNγ and TNFα co-expression in response to a polyclonal in vitro stimulation with PMA/Ionomycin mixture in comparison with the controls and COVID-19 convalescents.

During the acute phase of disease, the majority of SARS-CoV-2-specific CD4+ T cells in COVID-19 convalescents exhibited CD45RA–CCR7+ central memory phenotype [91]. Longitudinal analysis of S-specific Th cells showed that in convalescent peripheral blood, CD45RA–CCR7– Th cells were gradually replaced by CD45RA–CCR7+ central memory Th cells [92]. Next, Jung et al. revealed that among SARS-CoV-2-specific CD4+ T cells, the proportion of CM cells was rather stable and maintained at approximately 50% on average during 1 to 10 months PSO, while the level of CCR7−CD45RA− EM Th cells increased up to ~35% on average until 2 months PSO and then maintained thereafter [93]. Furthermore, the SARS-CoV-specific CD4+ T cells from the convalescent patients tended to exhibit CD27+CD45RO+ central memory phenotype with a higher frequency of polyfunctional memory cells producing IFNγ, TNFα, and IL-2 [94].

### 4.2. Polarization of T-helpers in COVID-19

#### 4.2.1. Th1 and Their Target Cells in COVID-19

When recognizing a specific antigen in peripheral tissues, effector Th1 cells are capable of producing IFNγ, which activates a wide range of immunocompetent cells, including CD8+ cytotoxic T lymphocytes, ILC1, macrophages, and B cells, which are involved in the elimination of intracellular pathogens (reviewed in [95]). The role of this Th subset in the pathogenesis of COVID-19 is controversial. Some studies pointed out a positive impact of IFNγ-producing Th1 cells in reducing this pathology and linked their increased activity with a milder course of the disease [96,97]. Moreover, when analyzing the pool of CD4 memory T cells, the cells that were capable of recognizing the epitopes of the three main SARS-CoV-2 structural proteins were predominantly from the Th1 subset [98,99], and this was noted already at very early stages after infection [91]. On the other hand, in aged individuals who are prone to a severe course of COVID-19, the levels of IFNγ-producing virus-specific cells decreased, which implied the importance of Th1 cells in the development of protective immune response [100].

However, the overproduction of IFNγ and TNFα by Th1 cells in response to SARS-CoV-2, as well as massive virus-infected cell death, can lead to lung injury and trigger acute respiratory distress syndrome. Thus, the migration of Th1 cells to inflamed tissues is indirectly indicated by a slight decrease in the proportion of these cells in peripheral blood at the acute phase of infection, which was observed in several independent studies [43,89,90]. Furthermore, patients with severe COVID-19 had a low frequency of Th1 [101]. In addition, some authors noted the accumulation of unconventional Th1 cells expressing CD161 and interleukin-1 receptor type I (IL-1RI) markers, which are more typical of Th17 cells, in the peripheral blood of COVID-19 patients with pneumonia, but unlike Th17 cells, these unconventional Th1 cells do not produce IL-17 [89].

The level of **cytotoxic CD8+ T-lymphocytes** in the peripheral blood of patients with COVID-19 seemed unchanged, but there was a decrease in ‘naïve’ CD3+CD8+ cells in circulation accompanied by an increased proportion of cells at the later stages of differentiation [38]. The authors noted that the proportion of mature perforin-positive cells within the total pool of CD8+ T cells grew as the severity of the disease increased. An increase in perforin+ cells notably correlated with a higher level of CRP in the serum. During the acute phase of SARS-CoV-2 infection, the percentage of CD8+ T cells in the peripheral blood reduced relative to the control [57]. Furthermore, the level of cytotoxic T-lymphocytes negatively correlated with the concentrations of D-dimer and ferritin in the serum of patients. Another group of researchers found a relationship between the level of HLA-DR-positive CD8+ T-lymphocytes and APACHE III scores, which made it possible to consider this cell population a promising marker of COVID-19 severity [27]. The minimum values of the relative and absolute numbers of CD3+CD8+ cells in peripheral blood were characteristic of patients with a poor outcome of COVID-19 [102].

Within the total pool of CD3+CD8+ cells (Figure 5), significant changes were identified in the ratio of different cell subsets at different stages of maturation. Thus, patients with COVID-19 showed a decrease in the proportion of EM1 cells with the CD45RA−CD27+CCR7− phenotype, while the relative count of EM2 and EMRA CD8+ T cells (with the CD45RA−CD27−CCR7+ and CD45RA+CD27−CCR7−phenotypes, respectively) significantly increased relative to the control values upon infection with SARS-CoV-2 [57]. The percentage and absolute count of CD39+ CD8+ T cells decreased, while PD-1-positive lymphocytes increased within the population of central memory cells (CD45RA-CD27+CCR7+) and EM1. Similar results were obtained by De Biasi et al., who reported a decrease in the proportion of ‘naive’ CD3+ CD8+ cells and central memory cells with the CCR7+CD45RA+CD28+CD27+ and CCR7−CD45RA+CD28+CD27+/− phenotypes, respectively [53]. All subsets of non-naive CD8+ T cells and the level of cells co-expressing CD38 and HLA-DR increased [57]. Moreover, patients with severe COVID-19 had higher counts of circulating activated CD38+HLADR+ and ‘exhausted’ CD27−CD28− CD8 phenotype cells compared to the mild severity group [101].

In addition to CD38 and HLA-DR as markers of chronic activation, an equally important prognostic value is attributed to the CD69 expression which is traditionally considered a marker of early activation of cytotoxic T cells. In all patients with COVID-19, CD3+CD8+CD69+ cell counts increased compared to the control; however, the concentration of these cells in the blood reached maximum values in patients with a poor prognosis of the disease [102]. As for cytotoxic CD8+ T cells, they increased the expression of PD-1 and TIM3, traditionally seen as markers of ‘cellular senescence’ [43] (Figure 4). In addition, the higher expression of PD-1 and TIM-3 by CD8+ T cells was closely associated with the severity of the disease, since the number of these cells in circulation in patients with severe COVID-19 exceeded the values obtained for patients with the mild disease [103]. The density of inhibitory molecules BTLA and TIGIT also increased on the surface of CD3+CD8+ cells in circulation, and these molecules can also be classified as factors limiting the expression of effector properties of cells [89]. Thus, high expression levels of effector molecules by CD8+ T cells in acute COVID-19 were associated with improved clinical outcomes, while the presence of ‘exhaustion’ molecules was associated with disease progression.

As for **NK cells**, numerous studies indicate significant changes in their functional properties in COVID-19. The very first studies showed a decrease in NK cells in circulating blood in patients with COVID-19 [104,105], and the minimum values were typical of critically ill patients [106,107,108]. A prolonged inflammatory process in COVID-19 associated with prolonged viral load was usually associated with a progressive decrease in NK cells in circulation and, hence, can be considered a marker of poor disease outcome [109]. An increase in CXCR3+ NK cells in the peripheral blood of critically ill patients was also noted, and their count decreased during effective therapy [110].

Firstly, NK cells were highly activated in COVID-19 patients and overexpressed CD25, CD69, and NKp44 on their surface [111]. However, the expression of the inhibitory receptor NKG2A increased [108,112], which is traditionally considered a marker of cellular senescence. Its presence is directly related to the impaired functional activity of NK cells, which is confirmed by lower levels of cytokine production (IFNγ, IL-2, and TNFα), as well as decreased expression of the degranulation marker CD107a by NK cells [112]. These results were confirmed in vitro: reduced production of IFNγ and CD107a by NK cells from COVID-19 patients was found when co-cultured with K562 cells [113]. Further studies on NK cells revealed overexpression of the three other markers of ‘cellular senescence’, LAG3, PDCD1, and HAVCR2 [28], as well as TIM-3 and PD-1 [102]. In addition, among the NK cells of patients with COVID-19, there was an increase in cells carrying CD39 on their surface. CD 39 is an cell-membrane enzyme capable of triggering a cascade of reactions leading to the formation of anti-inflammatory adenosine from proinflammatory ATP [108], which can also reduce the effectiveness of the antiviral response.

On the other hand, NK cells significantly increased the expression of the main activation markers Ki-67, HLA-DR, and CD69; however, no differences were noted between moderate and severe COVID-19 according to these parameters [113]. At the same time, the number of perforin-positive NK cells increased in the peripheral blood of patients [106]. The accumulation of mature NK cells in peripheral blood, which accumulated perforin or granzyme B in their cytoplasmic granules, correlated with the level of IL-6 in the serum. In addition, a high level of highly differentiated CD56dimCD57+ NK cells was closely associated with an unfavorable outcome of the disease and was characteristic of critically ill patients [102].

An inverse relationship was found between the serum concentrations of proinflammatory IL-6 and IL-8 and the level of perforin+ NK cells [114]. In addition, an increase in the levels of chemokines such as CCL3, CCL3, CCL4, CXCL9, CXCL10, and CXCL11 in BALF [115] can promote the directed migration of NK cells to the foci of viral infection [113], although the functional efficacy of NK cells circulating in the peripheral blood of patients with acute COVID-19 remains to be determined. Some data indicated a low number of mature CD16+CD57+ NK cells in BALF, whereas very high levels of these cells were found in the peripheral blood of patients [108].

Altogether, when analyzing the effector cells involved in the mediation of type 1 inflammatory response, it can be concluded that activated CD8+ T-lymphocytes and NK cells, which are released in circulation after differentiation in the lymphoid tissue, carry many markers of cellular aging on their surface and have reduced functional properties. This process is especially pronounced in patients with a severe course of COVID-19 and an unfavorable prognosis, which indicates low efficiency and numerous disruptions in the regulation of the response mediated by Th1 cells. Identifying these disruptions and analyzing their causes will promote a better understanding of the pathogenesis of COVID-19 and enhance the efficiency of treatment in patients with severe COVID-19.

#### 4.2.2. Th2 and Their Target Cells in COVID-19

Differentiated Th2 cells, when recognizing a pathogen, secrete cytokines IL-4, IL-5, and IL-13, thus initiating a type 2 immune response, which is associated with the activation of mast cells, basophils, and eosinophils, as well as epithelial cells capable of producing mucus [116]. The main target of Th2 cells is multicellular pathogens. However, in COVID-19, virus-specific Th2 cells were found [91,117], and high levels of Th2 cytokines were detected in the serum of patients at the acute phase of infection [90]. The proportion of T-helpers expressing CCR4 on their membrane and GATA3 in the nucleus was also higher in the peripheral blood of patients [53]. The increase in Th2 cells with the CXCR3–CCR6– phenotype in blood was closely associated with a poor outcome in patients with severe COVID-19, which made it possible to consider this indicator an independent prognostic marker [118] (Figure 4). An increased level of Th2 in blood and hyperactivation of these lymphocytes may be closely associated with such concomitant symptoms as intestinal hyperperistalsis, acidification of gastric juice, and shortness of breath, which, in this case, can be considered standard defense mechanisms to remove parasites by regulating Th2 cytokines [119]. As for the inflamed tissue, when analyzing the cells from the BALF of patients with severe COVID-19, upregulation of the genes encoding key factors responsible for the polarization of cells towards Th2 (GATA3, IL4R, and MAF) was found, although the production levels of the main Th2 cytokines did not differ between patients with different severities of COVID-19 [120]. Moreover, in patients who recovered from COVID-19, an elevated number of Th2 cells remained in the blood for several months, although the levels of IL-4, IL-5, and IL-13 did not differ significantly from the control values [121].

The main Th2 targets among immunocompetent cells are connective tissue mast cells and circulating basophils and eosinophils [122]. During the acute phase of SARS-CoV-2 infection, a decreased **basophil** count in the peripheral blood of patients was noted [123]. Importantly, the number of basophils was significantly reduced in patients with an unfavorable outcome relative to the values obtained for patients who recovered from COVID-19 [124]. Moreover, higher relative and absolute basophil numbers were found to negatively correlate with the risk of severe COVID-19 [125]. Thus, the restoration of the basophil count to the normal values could be regarded as a prognostic marker for the transition from the acute phase of inflammation caused by COVID-19 to the recovery phase [126].

As for alterations in the phenotype of basophils, a decrease in the expression levels of the integrin receptor CD11b and the receptor for prostaglandin D2 (CRTH2 or CD294) was found on the surface of basophils in patients with COVID-19 compared to cells of a similar population of apparently healthy volunteers [123]. When comparing patients with severe and mild COVID-19, an increased density of PDL1 expressed by basophils in severe patients was observed. Moreover, the density of PDL1 on basophils positively correlated with the severity of the disease (SOFA scores).

It is widely accepted that **mast cells** play a crucial role in protecting from helminths invasions during type 2 inflammation, although the role of these cells in antiviral protection is also crucial [127]. Their participation in COVID-19 pathogenesis may be associated with the release of various pro-inflammatory mediators, a high level of which can result in damage to lung tissue and activation of various immune and non-immune cells both in the inflammation site and at the systemic level. For example, the levels of mast cell-specific enzymes (chymase, β-tryptase, and carboxypeptidase A3) were increased in serum samples from patients with COVID-19 compared to those from healthy donors [128]. In addition, the analysis of lung tissue biopsy samples from patients with COVID-19 identified an increase in CD117+ mast cells and IL-4-expressing cells in the perivascular space and alveolar septa compared to controls [129]. A massive activation of mast cells, as well as their accumulation in the site of inflammation, make these cells a potential target for therapy in the acute course of COVID-19 [130]. The restriction or blockade of mast cell activation associated with the secretion of inflammatory mediators and the production of proinflammatory cytokines and chemokines can be used in clinical practice to reduce the volume of lung tissue damage [131].

Reduced levels of **eosinophils** in peripheral blood were typical for about 75% of patients with COVID-19 [132]. On admission to the hospital, patients with a low eosinophil count were more prone to fever, fatigue, and shortness of breath; they demonstrated lung tissue lesions during radiographic exacerbation more frequently and also stayed in the hospital for a longer time and had a more severe course of COVID-19 compared to hospitalized patients with the normal cell count in blood [133]. It is no less important to note that circulating eosinophils were significantly lower in critically ill COVID-19 patients compared to moderate or severe patients [134,135]. After controlling for confounding factors—age, gender, hypertension, coronary heart disease, diabetes, and chronic lung disease—a progressive decline in peripheral blood eosinophil levels was independently associated with a poor outcome. In addition, the concentration of eosinophils positively correlated with such important clinical indicators of COVID-19 severity as platelets frequency and D-dimer levels, while an inverse relationship was observed with the levels of urea, creatinine, aspartate aminotransferase, lactate dehydrogenase, and creatine kinase in serum. Still, some researchers have expressed doubts about the high predictive value of the association between the severe course of COVID-19 and a reduced circulating eosinophils count [125].

The eosinophil count in circulation upon infection with SARS-CoV-2 was reduced relative to the control, and the expression of CD294 on the surface of eosinophils was lower than the control values [123]. When comparing patients with mild and severe COVID-19, a higher density of PD-L1 expression by eosinophils of the latter was noted, which also positively correlated with the severity of the disease according to the WHO and SOFA scores.

A series of studies established the relationship between eosinophilia and the reduced severity of COVID-19, indicating the possibly important contribution of these cells to limiting inflammation during acute infection [136,137]. For instance, patients with eosinophilia had lower CRP levels, a milder clinical course, and better disease outcomes compared with patients without eosinophilia [138].

It can be assumed that the development of type 2 inflammation associated with an increase in Th2 cells and eosinophils in peripheral blood could be considered a favorable prognostic factor. There is evidence that Th2 cells and eosinophils can reduce the level of ACE2 expression on epithelial cells through the secretion of cytokines (primarily IL-13), and the epithelial cells are the gateway to SARS-CoV-2 [139], which is also confirmed by clinical observations of patients with respiratory diseases [140].

#### 4.2.3. Th17 and Their Target Cells in COVID-19

The most important role in the polarization of Th0 cells towards the Th17 phenotype is played by the proinflammatory cytokines IL-1β, IL-6, and IL-23 [95]. The levels of IL-1β and IL-6 particularly increase at the acute phase of the SARS-CoV-2 infection and could serve as additional markers of the disease severity [141]. The main Th17 effector cytokines are proteins of the IL-17 family (primarily IL-17A), which regulate the functions of neutrophils and their migration to the site of inflammation, and IL-22, the main function of which is to activate the protective functions of epithelial cells [116]. Both IL-17A [142] and IL-22 [143] can play a significant role in the COVID-19 pathogenesis and be considered therapeutic targets.

When analyzing the subset composition of Th17 cells in COVID-19, a decreased proportion of Th17.1 and Th1 lymphocytes capable of IFNγ production was noted, as well as a slight decrease in circulating Tregs [43] (Figure 4). In response to in vitro stimulation, T-helpers from patients infected with SARS-CoV-2 accumulated IL-17A and IL-2 more efficiently than the cells of a similar population from the control group [53]. The same study identified a decrease in the proportion of T-helpers expressing the key cell-surface Th17 antigens CD161 and CCR6, while the number of cells expressing Th2 markers (CCR4 and GATA3) was significantly higher than in the control. Similar results were obtained using molecular biological research methods: the expression of Th17-associated genes (RORC, IL17A, IL17F, and CCR6 decreased in the peripheral blood CD4+ T cells from patients with severe COVID-19 [120]. However, another study demonstrated an increased proportion of Th17 and follicular T cells alongside a slight decrease in Th1 in the peripheral blood of patients with COVID-19, while the values obtained for Th2 and Th17.1 did not differ from the control group [89] (Figure 4).

The lung tissues of patients with COVID-19 were enriched with CCR6 and IL-17A co-expressing cells, and high concentrations of IL-6, IL-17A, GM-CSF, and IFNγ were found in the liquid fraction of BALF [144]. It can be assumed that Th17 cells left the bloodstream at the acute phase of infection and migrated to the inflamed lung tissue, where they produced a wide range of proinflammatory cytokines capable of causing inflammation and damaging surrounding tissues through various effector mechanisms. The important role of Th17 in the pathogenesis of COVID-19 could be also evidenced by the fact that Th17 memory cells remained in circulation after the successful completion of the infectious process and elimination of the pathogen. For example, virus-specific Th17 memory cells capable of producing IL-17A, IL-17F, and IL-22 appear in response to in vitro stimulation by a pool of peptides derived from the S-protein [91]. Interestingly, robust Th17 responses were observed in patients with acute MERS-CoV and acute SARS-CoV infections [145,146], as well as a strong Th17 response was also observed in acute H1N1 influenza virus infection [147]. These data indicate that Th17 cells play a critical part in several viral infections, including SARS-CoV-2, MERS-CoV, SARS-CoV, and H1N1 and can play an important part in viral clearance, as well as a causal role in inflammation and tissue damage.

One of the potential markers of COVID-19 is the appearance of immature **neutrophils** with the CD16^low^CD11b^hi^ phenotype in the peripheral blood of patients with moderate to severe disease since these cells are detected in neither healthy volunteers nor patients with a mild form of the disease [38]. As the severity of the disease aggravates, the relative count of neutrophils with the CD16+CD11b^hi^ phenotype within the total CD45-positive leukocyte population increases. A lower proportion of these cells in patient follow-up is considered a favorable factor in COVID-19, while a further increase in the proportion of neutrophils within the total leukocytes in the peripheral blood is closely associated with an unfavorable outcome of the disease. On the other hand, the accumulation of immature monocytes with the CD10^Low^CD101–CXCR4+/− phenotype with pronounced suppressive properties was noted in the peripheral blood and lungs of patients with COVID-19 [47]. In addition, an increase in the proportion of CD10^Low^CD101+ neutrophils in peripheral blood was noted in patients with a mild course, while in severe COVID-19 cases, the CD10^Low^CD101– neutrophil subset was increased.

Patients with severe COVID-19 had a higher proportion of so-called low density neutrophils within the mononuclear cell fraction, including such immature neutrophil subsets as FUT4 (CD15)+CD63+CD66b+ pro-neutrophils and ITGAM (CD11b)+CD101+ pre-neutrophils [48]. These cells possessed a set of surface markers and an expression profile characteristic of granulocytic MDSCs with pronounced suppressive properties. At the same time, both mature and immature forms of neutrophils were characterized by a high expression of surface CD64 and PD-L1. Similarly, CD64 expression was higher in patients with severe and/or critical COVID-19 and further elevated in sepsis, along with increased PD-L1 expression [46]. If upon admission to the hospital a patient with COVID-19 had an elevated neutrophil count and an increased neutrophils/lymphocytes ratio alongside a reduced number of lymphocytes, eosinophils, and platelets, this patient was considered at risk, since such blood counts were characteristic of severe/critical and fatal patients [148]. Moreover, a prognosis was poor for patients with severe COVID-19 if they had a consistently high level or trend for an increase in neutrophil frequency and increased serum concentrations of IL-6, procalcitonin, D-dimer, SAA, and CRP during hospitalization. The results of the above study indicated that a progressive increase in the concentration of circulating neutrophils and basophils, as well as an increased level of IL-6 in peripheral blood, were associated with a lethal outcome. Other studies showed the importance of analyzing the neutrophils/CD3+CD8+ and neutrophils/lymphocytes ratios as predictors of severe COVID-19 when the values exceed 21.9 and 5.0, respectively [79]. A correlation was also found between the neutrophils/lymphocytes ratio and the severity of COVID-19 in APACHE III scores [27].

#### 4.2.4. Tfh Cells in COVID-19

The Tfh cells play a critical role in the maturation and differentiation of B cells as part of the germinal center reaction in the peripheral lymphoid tissue [149]. The circulating memory Tfh cell count seems to decrease in COVID-19 regardless of the severity of the disease [90] (Figure 4), although some studies found no difference between healthy volunteers and patients with COVID-19 [27] or even indicated an increase in the proportion of Tfh cells in circulation [89]. Interestingly, in the peripheral blood of patients recovered from COVID-19, the proportion of Tfh cells did not differ from the control values, while there was an increase in CXCR5+PD-1highCD4+ Tfh and CCR7loPD-1+ effector memory follicular cells (Tfh-em) and a decrease in CCR7hiPD-1– follicular cells of central memory (Tfh-cm) capable of migrating into lymphoid tissue [121].

At the same time, most studies noted changes in the subset’s composition of circulating memory Tfh cells. For instance, Mathew et al., 2020 revealed a significant increase in activated Tfh cells with the CD38+ICOS+ phenotype within the total pool of circulating memory Tfh cells (CD45RA-PD-1+CXCR5+). Another study showed that the proportion of PD-1+ICOS+ and CD38+HLA-DR+ cells increased within the CXCR5+CD4+ Tfh pool in the circulation in all patients with COVID-19 [87]. It should be noted that in the patients who had recovered from COVID-19, the level of activated Tfh was also significantly higher than that of the comparison group. In addition, more cells expressing Ki-67 and activation antigens CD38 and HLA-DR were detected among the total circulating Tfh cells of patients with COVID-19 than in the control group [57]. Higher concentrations of activated Tfh cells in the blood could be closely associated with an increase in circulating plasmablasts [27].

Despite their intensive activation, literature data indicate the low efficiency of Tfh cells in stimulating the humoral response. It is associated with the dysfunctional formation of germinal centers in B-dependent areas of lymph nodes, as well as with a decreased expression of the key transcription factor Bcl-6, which implements functional Tfh activities [54]. The results of histological studies indicate atrophy in the germinal centers of B-dependent areas of lymph nodes in the acute phase of the disease. According to Kaneko et al., this atrophy results from a ‘cytokine storm’, where TNFα, which can suppress Bcl-6+ Tfh cells differentiation, plays an important role.

In the peripheral blood of patients recovered from COVID-19, the proportion of CXCR3+CCR6– Tfh1 and CXCR3–CCR6– Tfh2 cells increased compared to the control, while the CXCR3–CCR6+ Tfh17 cell count was significantly reduced [121] (Figure 4). Recovered patients also showed a decrease in the circulating CD45RA–CD127–CD25+ CXCR5hiPD-1hi regulatory Tfhs in comparison with healthy volunteers. Furthermore, the proportion of circulating central memory Tfh cells was decreased in patients with COVID-19, while Tfh17 cells represented the most predominant subset in severe cases [58]. Patients with COVID-19 had circulating virus-specific CD45RA-CXCR5+ Tfhs capable of recognizing the S-protein, while the proportion of RBD-specific Tfhs was extremely low [150]. Moreover, the vast majority of SARS-CoV-2-specific Tfh cells were CCR6+CXCR3– Tfh17, but some of the cells had the Tfh1 phenotype (CCR6–CXCR3+). Recovered patients, whose plasma had a high neutralizing ability, showed higher counts of cTfh1 and cTfh2 cells, high levels of which positively correlated with the plasma neutralizing activity of blood serum in recovered subjects [150]. Finally, the subjects who recovered after COVID-19 showed increased frequencies of all peripheral blood effector Tfh cell subsets, primarily Tfh2 and Tfh17 cells [59]. Therefore, effective Tfh cell functioning and the proper balance between regulatory and pro-inflammatory Tfh cell subsets are crucial for the development of a functional humoral immune response with an appropriate quantity and quality of SARS-CoV-2-specific B cell memory.

## 5. Perspectives

COVID-19 infection can evidently lead to long-term disease in a significant proportion of survivors. The long-term consequences of COVID-19 and the post-acute COVID-19 syndrome, sometimes named ‘long COVID’ or ‘chronic COVID’, are still poorly understood but appear to affect a broad spectrum of patients, from ones who developed asymptomatic or mild acute COVID to those who required hospitalization and even received treatment in intensive care units [151,152,153]. Furthermore, in convalescent COVID-19 patients residual, SARS-CoV-2 viral antigens were detected in the gastrointestinal tract [154,155], suggesting that prolonged or latent chronic SARS-CoV-2 infection could be one of the causes of long-term COVID-19 manifestations.

For instance, platelet- and leukocyte-derived extracellular vesicles continue to increase even 1 month post-acute COVID-19, indicating that cellular activation persists long after the acute phase of SARS-CoV-2 infection [10]. Long-term alterations are found not only in primary B and T cell functions in COVID-19 patients up to 6 months following hospital discharge [156] but also in the vast majority of CD4+ T cell and CD8+ T cell subsets that are critical for the control of intracellular and extracellular pathogen infections [157]. Long-term alterations in maturation and differentiation of NK and CD8+ T cells, linked with high expression of inhibitory receptors and markers of ‘cellular senescence’ on their surface, as well as the low efficiency of target cell elimination can dramatically influence the effectiveness of antitumor immunity [158,159,160]. Moreover, alterations in both inflammation and immune responses in the setting of convalescent COVID-19 may influence future risk of various conditions such as cardiovascular, autoimmune, pulmonary, and neurological diseases, etc. The long-term consequences of COVID-19 may involve different cardiac pathophysiological mechanisms, including severe hypoxia-mediated injury, thromboembolic disease, systemic inflammatory response, and direct myocardial inflammation [161]. Hyperactivation of Th17 and Tfh cells, changes in their subsets, the imbalance between ‘regulatory’ and ‘pro-inflammatory’ T cell subsets in general, and uncontrolled B cell activation and antibody production increase the risk of autoimmune inflammation and autoimmune-related manifestations [6,162].

Circulating SARS-CoV-2-specific memory B and T cells and neutralizing antibodies were present in the majority of convalescent patients around 15 months post-acute COVID-19, implying the effective immunological memory formation and long-lasting immune response, while the reduction in the specific T cell response, but not B cell memory, was observed 12–15 months post-infection [163]. The presence of both SARS-CoV-2-specific polyfunctional memory T cells producing IFNγ, IL-2, and TNFα simultaneously and CCR7+CD45RA+CD95+ TSCM cells in convalescent patients 10 months post-infection regardless of the disease severity indicated that SARS-CoV-2-specific T cell memory could be long-lasting and effective in rapid viral clearance in the case of re-infection, as well as it could protect patients from developing severe COVID-19 [93]. Compared to milder infections, severe SARS-CoV-2 infection was associated with an elevated antibody and memory B cell response, and SARS-CoV-2-specific plasma cells were stably maintained in the bone marrow between 7 and 11 months after infection, supporting antibody levels in serum for a long time [164]. Notably, memory B cell response continued to undergo clonal evolution with an accumulation of somatic mutations and increasing affinity, which was overall consistent with the long-term persistence of viral antigens in germinal centers even 10–12 months post-symptom onset [165].

## 6. Conclusions

Although we have learnt a lot about SARS-CoV-2 virus and COVID-19 since its detection in the end of 2019, many questions still remain without clear answers. Less is known about the formation and maintenance of memory T and B cell subpopulations after the onset of symptoms, as well as about the influence of the dramatic alterations in T and B cell subsets, which were identified in the acute phase of SARS-CoV-2 infection, on long-lasting virus-specific memory. Another important question concerns the emergence of novel variants of SARS-CoV-2 that are potentially able to escape cellular and humoral immunity, both naturally acquired and/or induced by vaccination. However, SARS-CoV-2 immunology is changing swiftly. New research questions arise that focus on a more detailed characterization of the immune processes defining the acute phase of the SARS-CoV-2 infection and on developing a better understanding of the immunologic memory effectiveness and the long-term consequences of COVID-19.

## Figures and Tables

**Figure 1 viruses-14-01082-f001:**
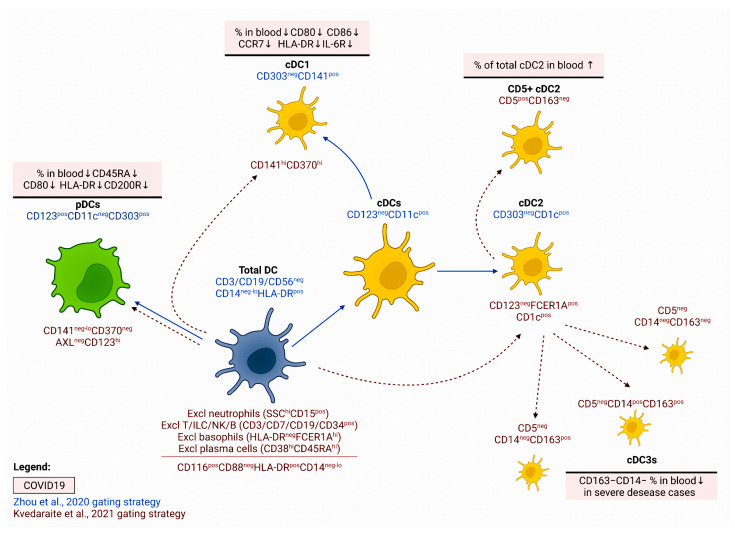
Altered surface phenotype of dendritic cells and their subsets in patients with COVID-19. DCs subsets: cDC1s carry BDCA-3 (CD141), Clec9A, CADM1, BTLA, and CD26 (are capable of cross-presentation of antigens to cytotoxic T-lymphocytes and polarization of ‘naive’ Th cells to Th1); cDC2s are CD1c+ (as well as FcεR1+SIRPA+; are capable to initiate responses mediated by various Th cell subsets) [30]; cDC2 can be subdivided into CD5+ DC2 and CD5− DC3 (consists of several subsets); finally, pDC are CD123+CD11c– and play a crucial role in antiviral response by secreting type I interferons and IL-12 (polarization of Th0 to Th1) [31]. Functional activity of almost all DC subsets can be reduced in severe COVID-19 cases (low levels of MHC molecules and limited costimulatory signaling (a decreased frequency of CD80 and CD86).

**Figure 2 viruses-14-01082-f002:**
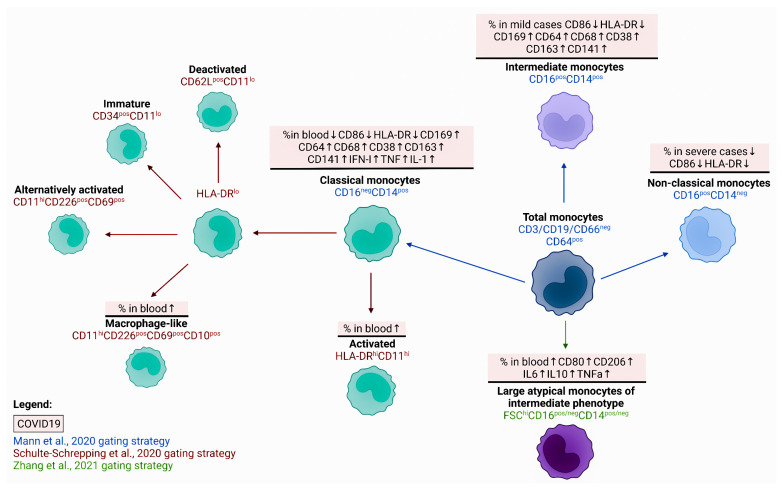
Major phenotypic changes within monocytes and their main subsets during acute COVID-19. Human peripheral blood monocytes are divided into three major populations: classical (CD14+CD16−), non-classical (CD14dimCD16+), and intermediate (CD14+CD16+). Currently, the data on monocyte subsets dynamics in COVID-19 are very contradictory, but all subsets were characterized by reduced expression of co-stimulatory and antigen-presenting molecules (CD86 and HLA-DR, respectively) and increased levels of different activation markers, including CD38, CD64, CD163, etc.

**Figure 3 viruses-14-01082-f003:**
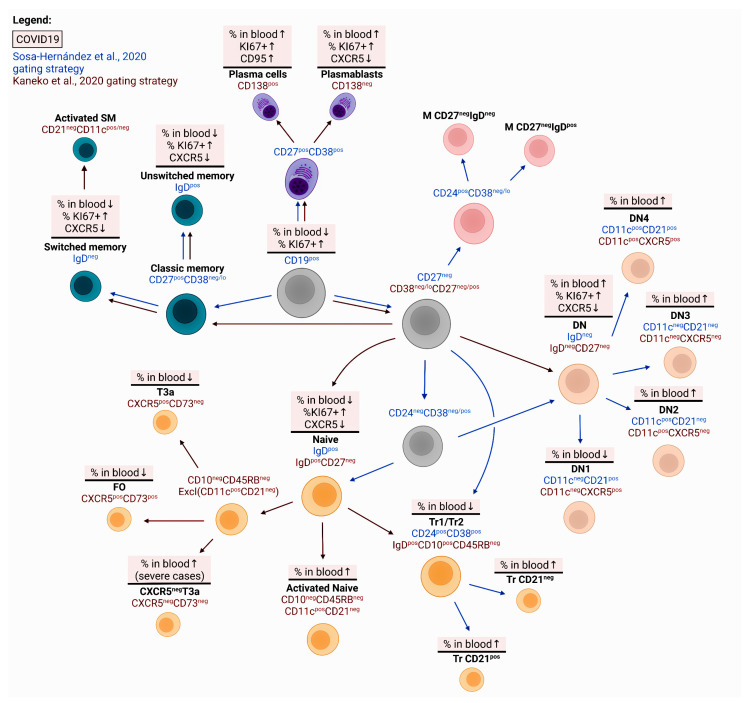
Abnormalities of B cell development and maturation in SARS-CoV-2-infected patient. Lower relative and absolute numbers of circulating main B cell subsets were found in patients with COVID-19 if compared to control, but these patients exhibited elevated levels of circulating CD20–CD38highCD27high plasmablasts and CD21-negative B lymphocytes pointing to impaired maturation and differentiation of memory and effector B cells in peripheral lymphoid tissue.

**Figure 4 viruses-14-01082-f004:**
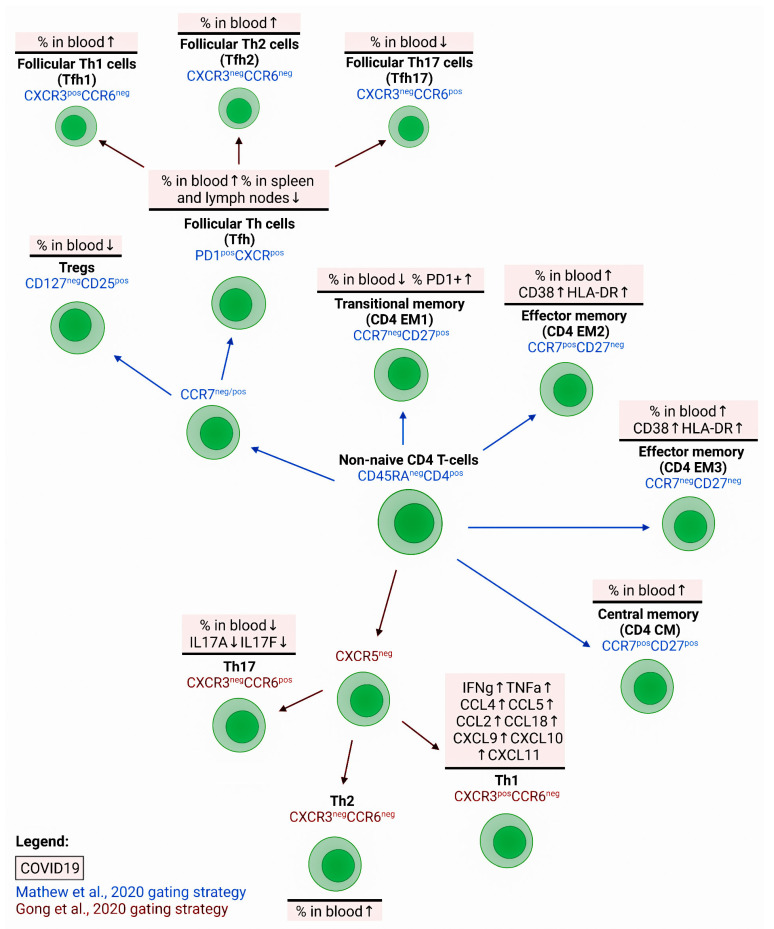
Impairment of peripheral blood Th cell subsets maturation and ‘polarization’ in patients with COVID-19. Abnormalities of Th cell maturation were linked with the decrease in ‘naïve’ CD4 T cells and increase of effector Th cell subsets. The relative number of activated (CD38+, HLA-DR+, or ICOS+) Th cell and their subsets were increased during the acute phase of SARS-CoV-2 infection. An imbalance of ‘polarized’ T cell subsets was frequently found in patients with COVID-19, being characterized by relatively high numbers of Th2 cells and low levels of Th17 and/or Tfh cells.

**Figure 5 viruses-14-01082-f005:**
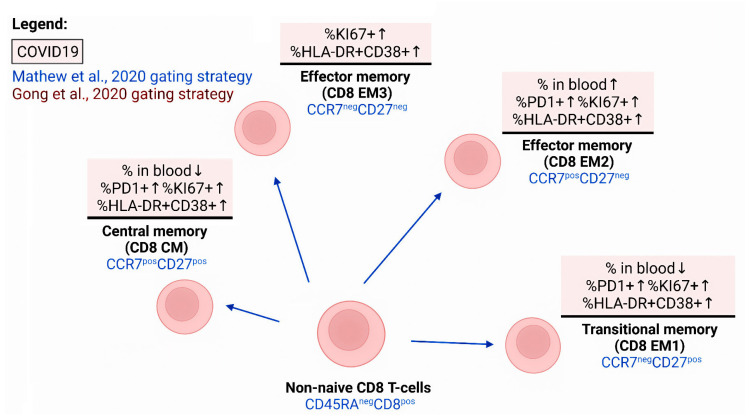
Impairment of peripheral blood CD8+ T cell subsets maturation in patients with COVID-19. Abnormalities of cytotoxic T cell maturation were linked with the increase in effector Th cell subsets. The relative number of activated (CD38+ and HLA-DR+) and ‘enhanced’ (PD-1+ and TIM-3+) cells were increased during the acute phase of SARS-CoV-2 infection.

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
