# Peer review of "Dysregulated Immune Responses in SARS-CoV-2-Infected Patients: A Comprehensive Overview"

_viruses, 2022, doi:10.3390/v14051082_

Round 1

Reviewer 1 Report

XCVXCV

The review "Dysregulated immune responses in SARS-CoV-2–infected patients: a comprehensive overview" provides a thorough analysis of data generated on pathological changes in T and B cell subsets and their phenotypes, accompanying the acute phase of the SARS-CoV-2 infection. This paper is completely within the scope of General Virology and it also responds pretty well to some of the topics of interest of the special issue "Viruses Research in Russia 2022". This kind of review is justifiable now, because it might help reveal new biomarkers that can be utilized to recognize case severity early as well as to provide additional objective information on the effective formation of SARS-CoV-2-specific immunity and predict long-term complications of COVID-19.  In the Referenses section, there are 7 self-citations by authors to published articles: five articles are experimental and two articles are reviews on the topic of the pathogenesis of COVID-19.

The present study is very interesting, but could be improved through aspect listed below.

When analyzing the  SARS-COV-2 specific memory cells, it would be interesting to  compare (briefly) with the formation of specific memory cell  to the previous coronavirus SARC-COV.

Author Response

Reviewer #1

The review "Dysregulated immune responses in SARS-CoV-2–infected patients: a comprehensive overview" provides a thorough analysis of data generated on pathological changes in T and B cell subsets and their phenotypes, accompanying the acute phase of the SARS-CoV-2 infection. This paper is completely within the scope of General Virology and it also responds pretty well to some of the topics of interest of the special issue "Viruses Research in Russia 2022". This kind of review is justifiable now, because it might help reveal new biomarkers that can be utilized to recognize case severity early as well as to provide additional objective information on the effective formation of SARS-CoV-2-specific immunity and predict long-term complications of COVID-19.  In the Referenses section, there are 7 self-citations by authors to published articles: five articles are experimental and two articles are reviews on the topic of the pathogenesis of COVID-19.

We thank the Reviewer#1 for finding our study interesting and for valuable comments.

The present study is very interesting, but could be improved through aspect listed below.

Question:

When analyzing the SARS-COV-2 specific memory cells, it would be interesting to compare (briefly) with the formation of specific memory cell  to the previous coronavirus SARC-COV.

Authors’ response:

We thank the Reviewer#1 for bringing up this important issue. Indeed, we were interested to describe in more details the role of antigen-specific T cells during SARS infection. The data on immune memory to the SARS and MERS viruses are still limited, but we were able to add several fragments dedicated to virus-specific memory Th cell phenotypes as well as virus-specific memory Th cell ‘polarization’.

Reviewer 2 Report

In general, this review sounds interesting, and addresses many important facets of the complex interaction between the immune response and the different clinical courses of SARS-CoV2.

The further critical points remain:

  1. The review is very difficult to read and contains redundant sections. I strongly recommend shortening the whole manuscript and keep it more concise without unnecessary repeats.
  2. The figures are confusing and need a careful revision to achieve more clarity. The figure legends must explain more in detail the main message of the depicted cell pictures. Please, use better color patterns to ease orientation for the reader.   
  3. The authors mention a central role of interleukin-6 in severe cases of COVID-19 what is right. Furthermore, they discuss potential positive effects of IL-6 blocking agents. However, they do not mention and discuss why IL-6 blockade has been not successful in severe COVID-19 (e.g.PMID: 32758889)
  4. As the review comes to Long-COVID, they do not cite and discuss new data showing that SRAS-CoV2 persist in intestinal villi spaces where it causes smoldering infection.
  5. The conclusion is too long, and again difficult to read. Shorten it and get more the point for a take home message.

Author Response

Reviewer #2

In general, this review sounds interesting, and addresses many important facets of the complex interaction between the immune response and the different clinical courses of SARS-CoV2.

We thank the Reviewer#2 for finding our study interesting and for valuable comments.

The further critical points remain:

  1. Question:

The review is very difficult to read and contains redundant sections. I strongly recommend shortening the whole manuscript and keep it more concise without unnecessary repeats.

Authors’ response:  

We thank the Reviewer#2 for this comment. To be clearer and in order to make the text more reader friendly, we deleted the redundant sections as well as the extra information that was not linked with peripheral blood cell phenotype changes during the acute SARS-CoV-2 infection. We also tried to make the text shorter.

  1. Question:

The figures are confusing and need a careful revision to achieve more clarity. The figure legends must explain more in detail the main message of the depicted cell pictures. Please, use better color patterns to ease orientation for the reader.  

Authors’ response:  

We highly appreciate this important suggestion of Reviewer#2 and have now modified the figures and their legend. Firstly, changed the color patterns in order to make the figures more colorful and to make cell subsets more different from each other. Secondly, we divided Figure 4 into two parts – now Figure 4 contains the information about alteration in Th cell phenotypes and subsets, while Figure 5 illustrates the changes in CD8+ T cells phenotypes. Finally, we added more detailed legends to all our Figures that summarized the main changes in cell subsets.

  1. Question:

The authors mention a central role of interleukin-6 in severe cases of COVID-19 what is right. Furthermore, they discuss potential positive effects of IL-6 blocking agents. However, they do not mention and discuss why IL-6 blockade has been not successful in severe COVID-19 (e.g. PMID: 32758889)

Authors’ response:  

We thank the Reviewer#2 for this comment. In the current study we mainly focused on peripheral blood cell populations, and we added the following text to your review:

However, IL-6 blockade may not be effective in some severe COVID-19 [83], and largely depended on many factors including baseline IL-6 levels, PaO2/FIO2, requirement of HFO or NIV, levels of CRP, ferritin, D-dimer, and LDH [84,85].

  1. Question:

As the review comes to Long-COVID, they do not cite and discuss new data showing that SARS-CoV-2 persist in intestinal villi spaces where it causes smoldering infection.

Authors’ response:

We understand the concern of the Reviewer #2 about the influence of long-term SARS-CoV-2 persistence on post-COVID, and we added two extra references to discuss this interesting question:

Furthermore, in convalescent COVID-19 patients residual SARS-CoV-2 viral antigens were detected in the gastrointestinal tract [154,155], suggesting that prolonged or latent chronic SARS-CoV-2 infection could be one of the causes of long-term COVID manifestations.

  1. Question:

The conclusion is too long, and again difficult to read. Shorten it and get more the point for a take home message.

Response:

We thank for this important suggestion. We divided the text into two part – ‘Perspectives’ and short ‘Conclusion’ – as well as we made modifications in both part, Please, see in the modified file.

Round 2

Reviewer 2 Report

The manuscript has been substantially improved. The review is more clearly arranged.